# Nomogram-Based Prediction of Survival in Stage IV Nasopharyngeal Carcinoma: A Retrospective Single-Center Study

**DOI:** 10.3390/diagnostics15111309

**Published:** 2025-05-23

**Authors:** Peng Yeh, Chih-Ming Chang, Li-Jen Liao, Chia-Yun Wu, Chen-Hsi Hsieh, Pei-Wei Shueng, Po-Wen Cheng, Wu-Chia Lo

**Affiliations:** 1Department of Otolaryngology, Far Eastern Memorial Hospital, New Taipei City 220216, Taiwan; 2Department of Biomedical Engineering, National Yang Ming Chiao Tung University, Taipei City 10044, Taiwan; 3Head and Neck Cancer Surveillance and Research Study Group, Far Eastern Memorial Hospital, New Taipei City 220216, Taiwan; 4Department of Electrical Engineering, Yuan Ze University, Taoyuan City 320315, Taiwan; 5Department of Oncology and Hematology, Far Eastern Memorial Hospital, New Taipei City 220216, Taiwan; 6Division of Radiation Oncology, Department of Radiology, Far Eastern Memorial Hospital, New Taipei City 220216, Taiwan; 7Department of Medicine, School of Medicine, National Yang Ming Chiao Tung University, Taipei City 10044, Taiwan; 8Graduate Institute of Medicine, Yuan Ze University, Taoyuan City 320315, Taiwan

**Keywords:** nasopharyngeal carcinoma (NPC), nomogram, body mass index (BMI), systemic inflammation response index (SIRI), systemic immune inflammation (SII)

## Abstract

**Background/Objectives**: To assess the pretreatment and posttreatment clinical factors associated with the rate of survival at 1, 3, and 5 years in stage IV nasopharyngeal carcinoma (NPC) patients. **Methods**: Clinicopathological characteristics of 61 Stage IV NPC patients diagnosed between 2008 and 2022 in a single tertiary medical center were retrospectively reviewed. Univariate and multivariate analyses were performed to evaluate the prognostic factors associated with overall survival (OS), disease-specific survival (DSS), and disease-free survival (DFS). A nomogram was developed to forecast DSS. **Results**: The OS at 1-year, 3-year, and 5-year were 93%, 70%, and 57%, while the DSS at 1-year, 3-year, and 5-year were 93%, 73%, and 58%, whereas the DFS at 1-year, 3-year, and 5-year were 51%, 44%, and 41%, respectively. In multivariate analyses, posttreatment body mass index (BMI) < 21.6 kg/m^2^ (hazard ratio [HR] 2.717, 95% confidence interval [CI] 1.248–5.917, *p* = 0.012) was an independent indicator for worsened OS. Posttreatment BMI < 21.6 kg/m^2^ (HR 3.003, 95% CI 1.340–6.757, *p* = 0.008) and pretreatment systemic inflammation response index (SIRI) ≥ 125 (HR 2.841, 95% CI 1.256–6.429, *p* = 0.012) were independent indicators for worsened DSS. Posttreatment BMI < 21.6 kg/m^2^ (HR 3.650, 95% CI 1.757–7.576, *p* = 0.001), change in BMI < −1.93 kg/m^2^ (HR 3.731, 95% CI 1.642–8.475, *p* = 0.002), and pretreatment SIRI ≥ 125 (HR 3.541, 95% CI 1.717–7.304, *p* = 0.001) were independent indicators for worsened DFS. A nomogram was created to predict DSS using posttreatment BMI and pretreatment SIRI. **Conclusions**: Associations with survival were observed between posttreatment BMI and OS, DSS, and DFS; pretreatment SIRI and DSS/DFS; and changes in BMI and DFS among patients with stage IV NPC. The developed nomogram aids in survival prediction.

## 1. Introduction

On a global scale, the age-standardized incidence rate of nasopharyngeal carcinoma (NPC) is 1.3 (2.2 for men and 0.87 for women), while the age-standardized mortality rate is 0.77 (1.2 for men and 0.39 for women) according to the 2020 GLOBOCAN data [1]. However, there is an unbalanced ethnic and geographic distribution of NPC cases, with the highest incidence occurring in South-Eastern Asia. The age-standardized incidence rate and age-standardized mortality rate of NPC among South-East Asians increased to 4.7 and 3.1 respectively, much higher compared to the global population. The latest data from the Taiwan cancer registry revealed that the age-standardized incidence rate among Taiwanese is 6.75 in males and 1.94 in females, while the age-standardized mortality rate is 2.80 in males and 0.62 in females [2]. The development of NPC is closely related to environmental factors, diet, and genetic factors [3]. In Southeast Asia, consumption of salted fish has been shown to contribute to NPC, though recent studies suggested that it has less influence than previously estimated, especially for intake during adulthood [4]. Cigarette smoking, alcohol consumption, and exposure to nitrites have been shown to be associated with an increased risk of the disease in Taiwan [5]. Family history and oral hygiene also contribute to the development of NPC [6,7]. Epstein-Barr virus (EBV) infection is perhaps the most common causal agent of NPC. World Health Organization (WHO) type 2 and 3 NPC accounts for greater than 97% of all cases of NPC in Taiwan, whereas type 1 is more common in Western countries. WHO type 2 and 3 NPCs are predominantly associated with EBV infection. Pretreatment plasma EBV DNA level was identified as a significant, and negative prognostic factor [8].

Given the biological characteristics of NPC, including its high sensitivity to irradiation, effective disease management depends heavily on accurate staging and appropriate treatment planning. Cancer staging for NPC primarily follows the American Joint Committee on Cancer (AJCC) tumor-node-metastasis (TNM) system, while treatment strategies are guided by the National Comprehensive Cancer Network (NCCN) guidelines [9,10]. The NCCN recommends induction chemotherapy (IC) followed by concurrent chemoradiotherapy (CCRT) or CCRT followed by adjuvant chemotherapy for advanced NPCs (stage III to IVA). The prognosis of NPC largely depends on its classification based on the TNM staging system. However, the current anatomy-based TNM staging system is insufficient for predicting prognosis or treatment benefits as there is tumor heterogeneity [11]. Other proven prognostic factors include plasma EBV DNA levels, pretreatment neutrophil-to-lymphocyte ratio (NLR), hemoglobin (Hb) levels, blood type A, and pretreatment serum lactate dehydrogenase levels [5,8,12,13,14]. There is a need to incorporate other clinical factors and molecular biomarkers into the system for improved risk stratification and treatment decision-making. Moreover, most studies focused on pretreatment markers and neglected posttreatment markers. This is particularly relevant in advanced NPC, where therapeutic outcomes are closely tied to the patient’s response to chemoradiation.

The effect of chemoradiation on cancer cells is shaped by the tumor microenvironment (TME), which is influenced by factors such as the patient’s nutritional status and antitumor immunity. Patients with better nutritional status tend to have improved survival rates. A higher pretreatment body mass index (BMI) is linked to more favorable treatment outcomes in head and neck cancer (HNC) patients who underwent chemoradiation [15]. The inflammatory status of cancer tissue is pivotal to the initiation, progression, and metastasis of cancer cells. The host’s inflammatory response also plays a key role in defending against cancer. Lymphocytes, neutrophils, monocytes, and platelets all play roles in the inflammatory response around tumor tissues. Tumor-associated neutrophils contribute to tumor growth by influencing angiogenesis, extravasation, and progression [16,17]. Lymphocyte plays an antitumor role with its specific adaptive immune responses [18]. Platelets stimulate metastasis by increasing microvascular permeability and influence the sensitivity of chemotherapy and other targeted therapies in cancer patients [19]. Different monocyte subsets have opposing roles in pro- and antitumoral immunity [20]. The lymphocyte-to-monocyte ratio (LMR), the NLR, and the platelet-to-lymphocyte ratio (PLR) have been established as predictors of treatment outcomes in HNCs [21,22,23]. The systemic immune-inflammation index (SII) and systemic inflammation response index (SIRI) further integrated the biomarkers. The SII combines platelets, neutrophils, and lymphocytes, while the SIRI combines neutrophils, monocytes, and lymphocytes in peripheral blood [24,25,26]. Thus, nutritional and inflammatory biomarkers may have prognostic value in cancer treatment outcomes.

This retrospective study conducted at a single institution aimed to evaluate and understand the pretreatment and posttreatment prognostic markers associated with 1-, 3-, and 5-year survival rates in stage IV NPC patients treated with CCRT. A nomogram was developed to estimate the survival rates of these patients at 1, 3, and 5 years.

## 2. Materials and Methods

### 2.1. Ethical Considerations

The study received approval from the Institutional Review Board (Far Eastern Memorial Hospital 112176-E, approval date 13 October 2023) of a tertiary care facility. The Research Ethics Review Committee of Far Eastern Memorial Hospital (No. 112176-E) approved a waiver of informed consent for this retrospective, anonymous study. This study fully adheres to the Declaration of Helsinki.

### 2.2. Patients and Data Acquisition

We reviewed patients diagnosed with advanced NPCs (stage IVA) between July 2008 and February 2022 at a single tertiary center. Radiologists reviewed the computed tomography scan or magnetic resonance imaging results, and the patients were retrospectively re-staged according to the 8th edition of the AJCC staging manual [10]. All patients included in this study were diagnosed with WHO type 2 and 3 NPCs without evidence of distant metastasis.

We reviewed the medical charts of the patients and analyzed data from the institutional cancer registration system. The collected data includes patient demographics, local and regional disease status (tumor and nodal status), and patient survival. Most importantly, the pretreatment and posttreatment nutritional status and laboratory data were examined.

The study’s inclusion criteria were as follows: (1) pathologically confirmed NPC; (2) clinical stage IVA disease, as defined by the 8th edition of the AJCC TNM staging system; (3) treated with CCRT with/without IC; (4) had comprehensive data for both pretreatment and posttreatment phases; (5) did not have concurrent second primary cancers or a history of previous malignancies. The patient exclusion criteria were as follows: (1) evidence of a distal metastasis at diagnosis; (2) treated with modalities other than CCRT with/without IC; (3) incomplete pretreatment and posttreatment medical records; (4) patients lost to follow-up.

### 2.3. Nutritional Status

The investigated nutritional parameters include pretreatment, posttreatment, and the change in Hb, body weight, body height, and BMI.

### 2.4. Inflammatory Biomarkers

We analyzed five inflammatory indices, both pretreatment and posttreatment: (1) LMR, the ratio of lymphocytes to monocytes; (2) NLR, the ratio of neutrophils to lymphocytes; (3) PLR, the ratio of platelets to lymphocytes; (4) SII, calculated as neutrophils multiplied by platelets divided by lymphocytes; and (5) SIRI, calculated as neutrophils multiplied by monocytes divided by lymphocytes.

### 2.5. Statistical Analysis

Statistical analyses were conducted using Stata software, version 12.0 (Stata Corp. LP, College Station, TX, USA). Data are presented as percentages (%), means ± standard deviations (SD), 95% confidence intervals (CI), hazard ratios (HR), and medians with interquartile ranges (IQR), as applicable. Continuous variables were analyzed using a two-sided Student’s *t*-test, while categorical variables were assessed using Fisher’s exact test or the Chi-squared test, as appropriate. The primary outcomes of this study were overall survival (OS), disease-specific survival (DSS), and disease-free survival (DFS). OS was defined as the duration from treatment completion to death from any cause or the last recorded instance of the patient being alive. DSS was defined as the time from treatment completion to death specifically caused by NPC or the last recorded instance of the patient being alive. DFS was defined as the interval from treatment completion to cancer recurrence or the last recorded instance of the patient being alive without recurrence. OS, DSS, and DFS were tracked from the date of treatment completion to the date of recurrence, death, or the final visit. The optimal cutoff values for univariate analysis of continuous variables were determined using the Youden index derived from receiver operating characteristic (ROC) curve analysis, selecting the point with the highest accuracy for predicting recurrence or death. These thresholds were defined prior to multivariable analysis to avoid data leakage. Variables that were statistically significant (*p* < 0.05) in univariate analyses were included in multivariate Cox regression analyses, adjusted for age and gender, using a forward stepwise method to address multicollinearity among parameters. Survival curves were generated using the Kaplan-Meier method and compared using the log-rank test. A nomogram was developed using variables that were statistically significant (*p* < 0.05) in multivariate Cox regression analyses to predict DSS following treatment completion in this patient cohort. The c-index was used to evaluate the nomogram’s discriminative ability, where a c-index of 0.5 indicates no discrimination and a c-index of 1 represents perfect discrimination. The total nomogram points for each patient were calculated and used to stratify patients into two groups based on the optimal cutoff points, identified through ROC curve analysis for the highest accuracy in predicting DSS.

## 3. Results

A total of 61 eligible patients with stage IVA NPC were included in this study. Table 1 summarizes the clinical and pathological characteristics of the cohort. Of these, 55 were males and 6 were females, all meeting the inclusion and exclusion criteria. The mean age at diagnosis was 56.7 years (range 24–88). The diagnoses were made between July 2008 and February 2022, with a mean follow-up duration of 1716 days (range, 169–5281 days).

All patients had stage IVA disease and were treated with CCRT with/without IC. Disease persistence was observed in 28 patients (45.90%) following the completion of treatment. Recurrence occurred in 33 patients, with local recurrence observed in 12 patients (19.67%), neck recurrence in 9 patients (14.75%), and distant failures in 24 patients (39.34%). The 1-year, 3-year, and 5-year OS rates were 93%, 70%, and 57%, respectively (Figure 1A). The DSS rates at 1 year, 3 years, and 5 years were 93%, 73%, and 58%, respectively (Figure 1B). The DFS rates at the same intervals were 51%, 44%, and 41%, respectively (Figure 1C).

### 3.1. Univariate Analysis

In the univariate analysis, significant risk factors for OS included pretreatment BMI (*p* = 0.026), posttreatment BMI (*p* = 0.015), pretreatment LMR (*p* = 0.047), pretreatment NLR (*p* = 0.024), pretreatment SII (*p* = 0.016), and pretreatment SIRI (*p* = 0.036). For DSS, significant risk factors were pretreatment BMI (*p* = 0.047), posttreatment BMI (*p* = 0.008), nodal status (*p* = 0.042), pretreatment LMR (*p* = 0.029), pretreatment NLR (*p* = 0.014), pretreatment SII (*p* = 0.010), pretreatment SIRI (*p* = 0.009), and change in SIRI (*p* = 0.045) (Figure 2). Significant risk factors for DFS included post-treatment BMI (*p* = 0.002), change in BMI (*p* = 0.029), pretreatment NLR (*p* = 0.005), pretreatment SII (*p* = 0.004), pretreatment SIRI (*p* = 0.007), and change in SIRI (*p* = 0.027) (Table 2).

### 3.2. Multivariate Analysis

Multivariate analyses using a forward stepwise Cox regression model identified posttreatment BMI < 21.6 kg/m^2^ (HR 2.717, 95% CI 1.248–5.917, *p* = 0.012) as an independent predictor of worsened OS. Additionally, pretreatment SIRI ≥ 125 (HR 2.841, 95% CI 1.256–6.429, *p* = 0.012, Figure 3A) and posttreatment BMI < 21.6 kg/m^2^ (HR 3.003, 95% CI 1.340–6.757, *p* = 0.008, Figure 3B) were identified as independent predictors of worsened DSS. For DFS, independent predictors of poorer outcomes included post-treatment BMI < 21.6 kg/m^2^ (HR 3.650, 95% CI 1.757–7.576, *p* = 0.001), change in BMI < −1.93 kg/m^2^ (HR 3.731, 95% CI 1.642–8.475, *p* = 0.002), and pretreatment SIRI ≥ 125 (HR 3.541, 95% CI 1.717–7.304, *p* = 0.001) in stage IV NPC patients (Table 3).

### 3.3. Nomogram

The independent risk factors for DSS were utilized to develop a nomogram (Figure 4) to predict DSS in stage IV NPC patients treated with chemoradiation. This tool aids in estimating survival outcomes based on individual patient characteristics. The two factors—posttreatment BMI < 21.6 kg/m^2^ and pretreatment SIRI ≥ 125—were each assigned a weighted score reflecting their impact on survival. For example, pretreatment SIRI ≥ 125 was given a score of 100. The nomogram can be utilized as follows: (1) Identify the ‘Points’ value of each predictive variable; (2) Sum the points to get the total score; (3) Locate the total score on the “Total Points” axis and draw a vertical line downward to determine the DSS at various time points.

The internal validation of the model yielded a discriminative c-index of 0.6918 (95% CI 0.631–0.876). The nomogram serves as a tool to differentiate between patients with disease persistence or recurrence and those who remained disease-free (Table 4). Patients with a low nomogram score (<92.5) had 1-year, 3-year, and 5-year DSS rates of 80%, 46%, and 23%, respectively, compared to 97%, 82%, and 69% for those with a high nomogram score (≥92.5) (*p* < 0.0001; Figure 5). A nomogram score cutoff of 92.5 provided effective discrimination among patients with stage IVA NPC.

## 4. Discussion

In this study, we identified significant associations between survival outcomes and several factors: posttreatment BMI was linked to OS, DSS, and DFS; pretreatment SIRI was associated with DSS and DFS; and changes in BMI were related to DFS. These factors serve as independent prognostic parameters of 5-year survival in patients with stage IV NPC. Examining hemogram and nutritional status is cost-effective and easily accessible. Multidisciplinary treatment drafting for stage IV NPC includes detailed pretreatment evaluation, disease status determination, individualized treatment strategy, and posttreatment assessment. The latest NCCN guideline recommends comprehensive patient profiling, including monitoring nutritional status and hemograms before and after treatment [9].

Patients with a posttreatment BMI ≥ 21.6 kg/m^2^ had significantly better OS, DSS, and DFS compared to those with a BMI < 21.6 kg/m^2^. HNC patients with elevated BMI, or in other words overweight patients, have been found to have improved survival, lower disease-related mortality, and recurrence rate as compared to their normal or underweight counterparts, regardless of treatment strategy [15,27,28,29,30]. BMI before treatment is an independent prognostic factor for NPC survival [31,32]. However, our findings differ slightly from most studies as we found the significance of posttreatment BMI instead of pretreatment BMI on survival. Change in BMI < −1.93 kg/m^2^, which means a higher decrease in BMI as compared to their counterparts, has an effect on DFS but not on OS or DSS. Maintaining posttreatment nutritional status is important among advanced NPC patients.

Studies have demonstrated the associations between inflammatory status and HNC oncogenesis [16,17,18,19]. Tumor-associated neutrophils promote tumor growth through the release of reactive oxygen species, paracrine signaling, and alterations of TME while also participating in early antitumorigenic roles [33]. Lymphocyte infiltrating TME, especially Th1 and T regulatory subsets, appears to have tumor-suppressive functions against HNCs [18]. Circulating tumor cells activate platelets to create a supportive microenvironment, accelerate extravasation, and promote the establishment of micrometastatic foci [34]. Monocytes contribute to both pro- and antitumoral immunity [20].

A selected biomarker should be accessible and economical, yet at the same time provide high accuracy and precision to be an appropriate candidate. Peripheral blood biomarkers have been shown to be linked with the prognosis of HNC treatment. We collected platelet counts, as well as absolute neutrophil, lymphocyte, and monocyte counts. LMR, NLR, PLR, SII, and SIRI are aggregates of individual blood biomarkers that reflect the inflammatory interaction between cancer and immune cells. These combinations were developed in the hope of creating a novel systemic inflammatory response biomarker with high predictive value. The SIRI was initially developed in 2016 to predict the survival of pancreatic cancer patients who received chemotherapy [35]. Previous studies have demonstrated that a high SIRI is an independent poor prognostic factor for HNCs [26,36,37]. Elevated SIRI reflects an immune suppressive TME [38]. The 5-year disease-specific mortality risk in advanced NPC patients was significantly higher for those with a pretreatment SIRI ≥ 125 compared to their counterparts (HR 2.841, 95% CI 1.256–6.429, *p* = 0.012). In the high SIRI group, the survival rate decreased from 86% at 1 year to 62% at 3 years, and further to 45% at 5 years.

The underlying mechanism linking SIRI to prognosis in NPC, as well as its relationship with EBV, remains unclear. However, the prognostic value of SIRI may be attributed to its individual components. Neutrophils can promote immunosuppression by releasing cytokines such as interleukin-10 (IL-10) and transforming growth factor-β (TGF-β), which inhibit T cell proliferation and activation. In contrast, lymphocytes have antitumor activity, suppressing tumor growth and metastasis through the secretion of interferon-γ (IFN-γ) and tumor necrosis factor-α (TNF-α). Monocytes and monocyte-derived M2 macrophages contribute to tumor progression by facilitating growth, invasion, immune evasion, and metastasis [39].

We found that posttreatment BMI is associated with survival in patients with stage IV NPC, in contrast to the commonly reported association of pretreatment BMI with survival in other HNCs [15,40]. Patients with HNC often experience swallowing difficulties due to the nature of the disease’s location. Obstruction of the enteral route by advanced oral, oropharyngeal, or hypopharyngeal cancers leads to malnutrition prior to treatment. In contrast, the most significant weight loss typically occurs during the middle to late stages of CCRT in NPC, coinciding with an increased risk of malnutrition in the later phases of treatment. This suggests that the greatest cumulative nutritional loss occurs after treatment completion, likely due to treatment-related toxicities. However, previous studies have not explored the relationship between posttreatment nutritional status and survival in locally advanced NPC. Our findings build upon earlier research, demonstrating that posttreatment BMI is a distinct and meaningful prognostic factor in this context. Nevertheless, caution is warranted, as BMI may be influenced by cancer type, the timing of its assessment, and variations in nutritional support practices.

The observed decline in BMI and elevated SIRI may reflect treatment-induced cachexia, poor recovery, and overall cancer burden. Nutritional status is a multifactorial variable influenced by age and the timing of measurement [41]. Previous studies have shown that weight loss typically peaks during the middle to late phases of CCRT [40], underscoring the importance of active surveillance and timely intervention. However, cancer cachexia is a complex syndrome resulting from metabolic, immunological, and endocrine dysregulation, and nutritional supplementation alone is often insufficient to reverse it [42]. Therefore, the notion that improving nutritional status—and consequently increasing posttreatment BMI—can directly enhance cancer survival may be overly simplistic.

SIRI, which incorporates monocyte, neutrophil, and lymphocyte counts, serves as an integrated marker of chronic low-grade inflammation. These immune cells play complex and interrelated roles in both pro- and antitumor immunity. Whether SIRI is modifiable remains uncertain. In this study, changes in BMI, posttreatment BMI, and pretreatment SIRI appear to reflect a patient’s overall resilience to therapy. Further research is needed to clarify whether modifying nutritional and inflammatory statuses can translate into improved survival outcomes.

Subgroup analyses were conducted to compare changes in BMI, posttreatment BMI, and pretreatment SIRI between patients with and without recurrence or persistent disease. Patients experiencing recurrence or persistence had a higher proportion with a change in BMI of <−1.93 kg/m^2^, posttreatment BMI < 21.6 kg/m^2^, and pretreatment SIRI ≥ 125, compared to those without (Appendix A). These findings may reflect that patients with residual or recurrent disease have lower BMI and greater weight loss as a result of cancer progression or cachexia, rather than these being independent predictors of prognosis. Therefore, the possibility of reverse causation should be carefully considered in interpreting these variables as prognostic factors. Although SIRI was associated with survival outcomes in our study, it may also reflect underlying tumor burden or EBV-related inflammation rather than acting as an independent predictor. Notably, key prognostic factors such as EBV DNA levels or total tumor volume were not included in the current Cox regression analyses, which may confound the observed association. Therefore, the interpretation of SIRI should be made with caution, and further studies are warranted to determine its independent prognostic value.

For stage IV NPC patients, the current treatment algorithm recommends IC followed by CCRT, CCRT followed by adjuvant chemotherapy, or participation in clinical trials [9]. IC significantly enhances survival compared to CCRT alone for advanced-stage NPC [43]. Unfortunately, the treatment outcome of IC with CCRT remains poor for these patients. The other preferred treatment is a clinical trial. The NCCN advocates that the optimal management for any cancer patient is through clinical trials and encourages advanced-stage NPC patients to participate in them. The selection of poor responders is crucial in treatment decisions.

To our knowledge, this is the first study to develop a 5-year nomogram using both pretreatment and posttreatment markers to predict survival in stage IV NPC patients. Tumor heterogeneity necessitates individualized cancer therapy tailored for advanced disease. Nomograms are visual tools that help physicians easily identify potential poor responders. The 1-year, 3-year, and 5-year DSS rates for patients with a nomogram score of <92.5 were 80%, 46%, and 23%, respectively, compared to 97%, 82%, and 69% for those with a score of ≥92.5 (*p* < 0.0001). A cutoff nomogram score of 92.5 clearly differentiated the disease recurrence rates. Patients with a nomogram score of <92.5 had a 2.2-fold risk of disease recurrence compared to those with a score of ≥92.5 (100% versus 45.65%, *p* = 0.001). Patients with poor prognosis may be appropriate candidates for intensive nutritional support. Physicians should consider early enrollment of these individuals in clinical trials as an alternative to standard therapies.

Our study has some limitations. Firstly, this is a single-institution retrospective cohort study with a relatively small sample size. Due to its retrospective nature, confounding selection bias may be unavoidable. Practice patterns may differ among institutions due to variations in target cohorts. We adhered to the conventional guidelines when treating our patients, as most academic institutions do. Secondly, our cohort originates from a region where the disease is prevalent, so the findings may not be applicable to NPC populations worldwide. Thirdly, other carcinogenic factors like tobacco use, alcohol consumption, and existing comorbidities, may have impacted our patients’ survival rates. Moreover, the EBV status of our patients was incomplete, as EBV-DNA testing was not routinely implemented at our hospital until 2016. Fourthly, specific cutoff values were selected for BMI and SIRI to develop the nomogram. Results might differ using other cutoff points. Furthermore, despite the internal validation demonstrating excellent discriminative ability, there is a lack of external validation. The cutoff values derived from this single-institution cohort may have limited generalizability to other populations. Additional external validation by large cohorts is necessary. Fifth, systemic inflammation markers such as posttreatment SII and SIRI may be influenced by confounding factors, including infections. Additionally, BMI is a crude indicator of nutritional status; incorporating measures such as muscle mass or assessing for sarcopenia would provide a more accurate representation of a patient’s nutritional condition. Finally, the high male-to-female ratio in our cohort may differ significantly from other populations. Based on data from the Taiwan Cancer Registry Database, the nationwide male-to-female ratio for stage IV NPC without metastasis ranged from 2.67 to 3.94 between 2017 and 2021. The high male-to-female ratio reflects the demographics of local stage IV NPC patients. The strength of this study is the extensive pretreatment and posttreatment clinicopathological parameters of our cohort. This single tertiary center stage IV NPC cohort was unanimously treated with CCRT. The patients were observed over an extended period (mean, 1716 ± 1096 days). The biomarkers required for nomogram utilization have high accessibility.

## 5. Conclusions

In summary, the posttreatment BMI, change in BMI, and pretreatment SIRI are important prognostic features for advanced stage IV NPCs. Associations with survival were observed between posttreatment BMI and OS, DSS, and DFS; pretreatment SIRI and DSS/DFS; and changes in BMI and DFS among patients with stage IV NPC. Our developed nomogram may aid in the stratification of patients and individualized treatment strategy selection. Additional external validation and confirmation through large cohort prospective studies is necessary.

## Figures and Tables

**Figure 1 diagnostics-15-01309-f001:**
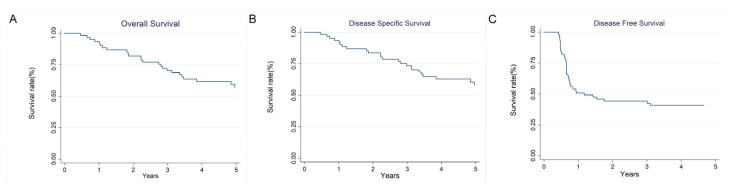
(**A**) Kaplan-Meier analyses of OS in stage IV NPC patients. The OS at 1-year, 3-year, and 5-year were 93%, 70%, and 57%, respectively. (**B**) Kaplan-Meier analyses of DSS in stage IV NPC patients. The DSS at 1-year, 3-year, and 5-year were 93%, 73%, and 58%, respectively. (**C**) Kaplan-Meier analyses of DFS in stage IV NPC patients. The DFS at 1-year, 3-year, and 5-year were 51%, 44%, and 41%, respectively.

**Figure 2 diagnostics-15-01309-f002:**
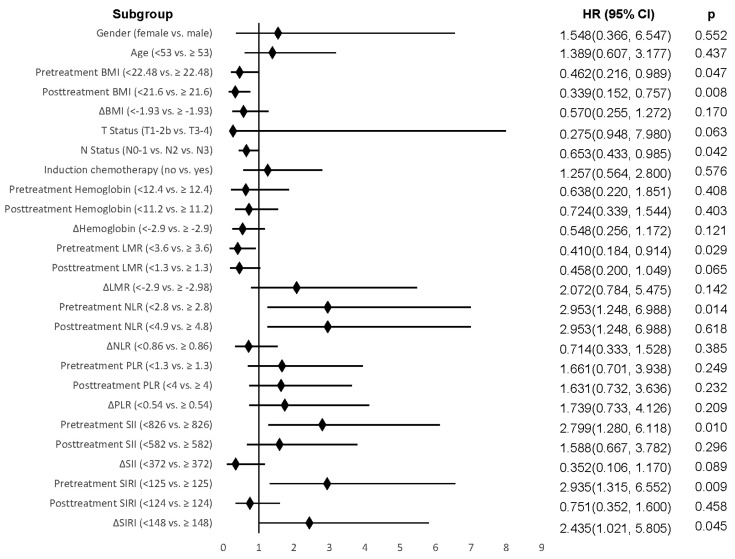
Forest plot of DSS with the variables included in the univariable analysis.

**Figure 3 diagnostics-15-01309-f003:**
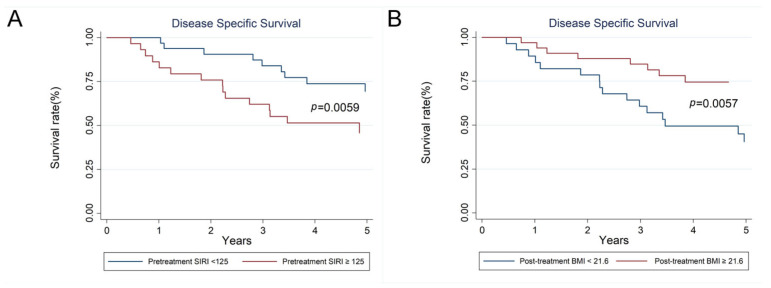
(**A**) The 1-year, 3-year, and 5-year DSS for stage IV NPC patients with pretreatment SIRI ≥ 125 versus pretreatment SIRI < 125 were 86%, 62%, and 45% versus 100%, 84%, and 69%, respectively. (**B**) The 1-year, 3-year, and 5-year DSS for stage IV NPC patients with posttreatment BMI ≥ 21.6 versus posttreatment BMI < 21.6 were 97%, 85%, and 74% versus 89%, 61%, and 41%, respectively.

**Figure 4 diagnostics-15-01309-f004:**
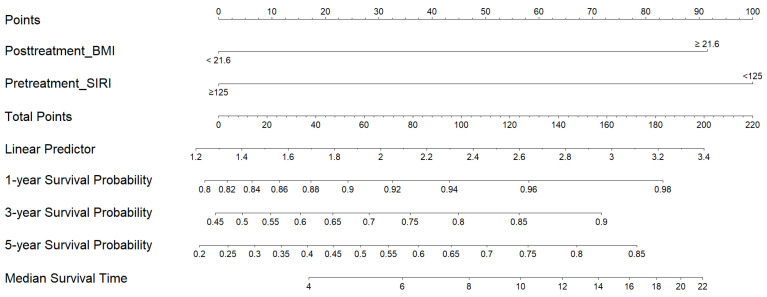
The nomogram for DSS of stage IV NPC patients. Instruction for usage: Find the respective value of each predictive variable by drawing a vertical line to the “Points” line. Repeat the process for each predictive variable. Sum the total points and draw a vertical line from the “Total points” line down to the 1-year, 3-year, 5-year, and median survival timelines to find the predicted DSS at different times.

**Figure 5 diagnostics-15-01309-f005:**
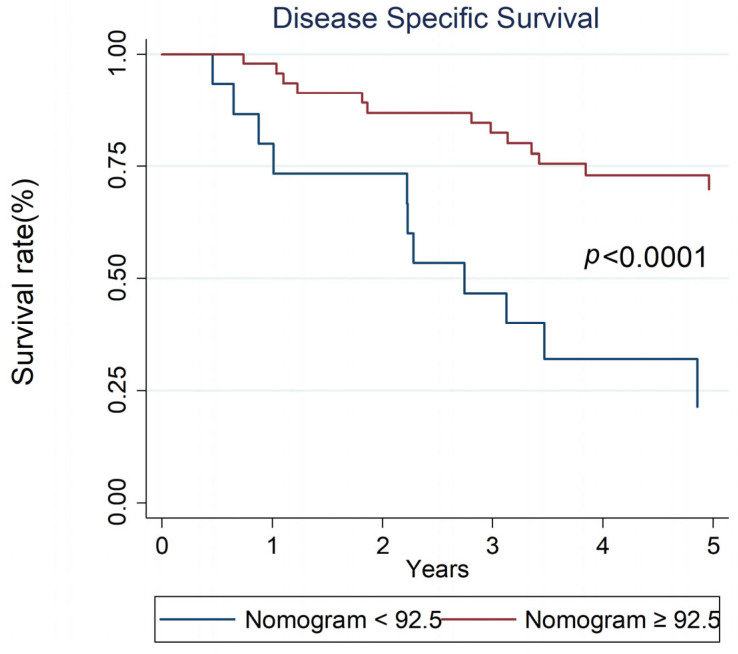
The 1-year, 3-year, and 5-year DSS for stage IV NPC patients with low nomogram score (<92.5) versus high nomogram score (≥92.5) were 80%, 46%, and 23% versus 97%, 82%, and 69% (*p* < 0.0001), respectively.

**Table 1 diagnostics-15-01309-t001:** Characteristics and clinicopathological details of the recruited patients.

		*N* (%) or Mean ± SD or Median (Range, IQR)
Age (years)		56.7 ± 11 (24–88)
Gender	Male	55 (90.16%)
	Female	6 (9.84%)
Clinical T Classification	T1/T2/T3/T4	12 (19.67%)/7 (18.03%)/12 (19.67%)/30 (49.18%)
Clinical N Classification	N0/N1/N2/N3	5 (8.20%)/12 (19.67%)/9 (14.75%)/20 (32.79%)
Body Height (cm)		166.7 ± 7.4 (149–189)
Pretreatment BW (kg)		69.6 ± 16.5 (43–146)
Posttreatment BW (kg)		62.2 ± 11.9 (41–107)
Pretreatment BMI (kg/m^2^)		24.9 ± 4.5 (16.38–40.87)
Posttreatment BMI (kg/m^2^)		62.2 ± 11.9 (41–107)
Induction chemotherapy	With/Without	37 (60.66%)/24 (39.34%)
Death within 5 years		28 (45.90%)
Disease Persistence		28 (45.90%)
Recurrence	Local/Regional/Distant	12 (19.67%)/9 (14.75%)/24 (39.34%)
Follow-up duration (days)		1716 ± 1096 (169–5281)
Smoking status	Yes/No	34 (55.74%)/27 (44.26%)
ECOG performance status	0/1/2/4	30 (49.18%)/29 (47.54%)/1 (1.64%)/1 (1.64%)
Hypertension	Yes/No	5 (8.20%)/56 (91.80%)
Diabetes	Yes/No	2 (3.28%)/59 (96.72%)
Hepatitis B	Yes/No	3 (4.92%)/58 (95.08%)
Coronary artery disease	Yes/No	2 (3.28%)/59 (96.72%)

BW: body weight; BMI: body mass index; ECOG: Eastern Cooperative Oncology Group.

**Table 2 diagnostics-15-01309-t002:** Univariate analyses of clinicopathological factors that related to overall survival (OS), disease-specific survival (DSS), and disease-free survival (DFS) in stage IV nasopharyngeal carcinoma patients.

Variables	Dichotomized Units	OS				DSS				DFS			
		HR	95%CI		*p*-Value	HR	95%CI		*p*-Value	HR	95%CI		*p*-Value
Gender	Female vs. male	1.611	0.382	6.800	0.516	1.548	0.366	6.547	0.552	3.083	1.346	0.413	4.389
Age	<53 vs. ≥53	1.747	0.742	4.116	0.202	1.389	0.607	3.177	0.437	1.423	0.711	2.849	0.319
Pretreatment BMI (kg/m^2^)	<22.48 vs. ≥22.48	0.430	0.204	0.906	0.026	0.462	0.216	0.989	0.047	0.533	0.272	1.043	0.066
Posttreatment BMI (kg/m^2^)	<21.6 vs. ≥21.6	0.380	0.175	0.826	0.015	0.339	0.152	0.757	0.008	0.353	0.180	0.693	0.002
Δ BMI (kg/m^2^)	<−1.93 vs. ≥−1.93	0.634	0.292	1.378	0.250	0.570	0.255	1.272	0.170	0.443	0.213	0.921	0.029
T status	T1&T2 vs. T3&T4	2.880	0.996	8.328	0.051	0.275	0.948	7.980	0.063	1.657	0.776	3.536	0.192
N status	N0-1 vs. N2 vs. N3	0.681	0.445	1.019	0.062	0.653	0.433	0.985	0.042	0.723	0.502	1.040	0.080
Pretreatment Hb (gm/dL)	<16 vs. ≥16	0.513	0.194	1.353	0.117	0.638	0.220	1.851	0.408	0.800	0.311	2.060	0.644
Posttreatment Hb (gm/dL)	<11.2 vs. ≥11.2	0.582	0.276	1.225	0.154	0.724	0.339	1.544	0.403	0.948	0.488	1.842	0.875
Δ Hb (gm/dL)	<−2.9 vs. ≥−2.9	0.684	0.326	1.436	0.316	0.548	0.256	1.172	0.121	0.598	0.310	1.154	0.125
Pretreatment LMR	<3.6 vs. ≥3.6	0.456	0.210	0.990	0.047	0.410	0.184	0.914	0.029	0.553	0.286	1.070	0.078
Posttreatment LMR	<1.3 vs. ≥1.3	0.485	0.213	1.104	0.085	0.458	0.200	1.049	0.065	0.493	0.231	1.054	0.068
Δ LMR	<−2.98 vs. ≥−2.98	1.414	0.601	3.327	0.728	2.072	0.784	5.475	0.142	1.369	0.659	2.843	0.400
Pretreatment PLR	<1.3 vs. ≥1.3	1.748	0.741	4.210	0.202	1.661	0.701	3.938	0.249	1.362	0.680	2.728	0.383
Posttreatment PLR	<4 vs. ≥4	1.557	0.704	3.446	0.275	1.631	0.732	3.636	0.232	1.892	0.944	3.794	0.072
Δ PLR	<0.54 vs. ≥0.54	1.829	0.775	4.314	0.168	1.739	0.733	4.126	0.209	1.856	0.843	4.083	0.124
Pretreatment NLR	<2.8 vs. ≥2.8	2.575	1.134	5.851	0.024	2.953	1.248	6.988	0.014	2.725	1.355	5.481	0.005
Posttreatment NLR	<4.8 vs. ≥4.8	1.312	0.622	2.767	0.476	2.953	1.248	6.988	0.618	1.881	0.973	3.638	0.060
Δ NLR	<0.86 vs. ≥0.86	0.771	0.366	1.625	0.495	0.714	0.333	1.528	0.385	0.809	0.420	1.559	0.527
Pretreatment SII	<826 vs. ≥826	2.533	1.185	5.412	0.016	2.799	1.280	6.118	0.010	2.699	1.385	5.257	0.004
Posttreatment SII	<582 vs. ≥582	1.655	0.699	3.920	0.252	1.588	0.667	3.782	0.296	2.052	0.897	4.692	0.089
Δ SII	<372 vs. ≥372	0.340	0.103	1.127	0.078	0.352	0.106	1.170	0.089	0.636	0.277	1.457	0.284
Pretreatment SIRI	<125 vs. ≥125	2.260	1.056	4.838	0.036	2.935	1.315	6.552	0.009	2.558	1.297	5.044	0.007
Posttreatment SIRI	<124 vs. ≥124	0.804	0.382	1.691	0.565	0.751	0.352	1.600	0.458	0.980	0.501	1.916	0.953
Δ SIRI	<148 vs. ≥148	2.299	0.970	5.449	0.059	2.435	1.021	5.805	0.045	2.451	1.108	5.425	0.027
Induction chemotherapy	No vs. Yes	1.134	0.523	2.459	0.751	1.257	0.564	2.800	0.576	0.775	0.398	1.507	0.452
ECOG performance status	0–1 vs. 2–4	1.703	0.403	7.200	0.469	1.759	0.415	7.453	0.443	2.174	0.513	9.214	0.292

BH: body height; BW: body weight; BMI: body mass index; Hb: hemoglobin; Δ: change in the variable; LMR: lymphocyte/monocyte ratio; PLR: platelet/lymphocyte ratio; NLR: neutrophil/lymphocyte ratio; SII: systemic immune inflammation index; SIRI: systemic inflammation response index; ECOG: Eastern Cooperative Oncology Group.

**Table 3 diagnostics-15-01309-t003:** Multivariate logistic regression analyses of clinicopathological factors that related to overall survival (OS), disease-specific survival (DSS), and disease-free survival (DFS) in stage IV nasopharyngeal carcinoma patients (only presenting variables with statistical significance).

Variables	Dichotomized Units	OS				DSS				DFS			
		HR	95%CI		*p*-Value	HR	95%CI		*p*-Value	HR	95%CI		*p*-Value
Posttreatment BMI (kg/m^2^)	<21.6 vs. ≥21.6	0.368	0.169	0.801	0.012	0.333	0.148	0.746	0.008	0.274	0.132	0.569	0.001
Δ BMI (kg/m^2^)	<−1.93 vs. ≥−1.93									0.268	0.118	0.609	0.002
Pretreatment SIRI	<125 vs. ≥125					2.841	1.256	6.429	0.012	3.541	1.717	7.304	0.001

BMI: body mass index; Δ: change in the variable; SIRI: systemic inflammation response index.

**Table 4 diagnostics-15-01309-t004:** Rate of disease persistence/recurrence according to the DSS nomogram score in patients with stage IV nasopharyngeal carcinoma.

		Disease Persistence/Recurrence (*n* = 36)	None (*n* = 25)	*p*-Value
Nomogram score	<92.5	15 (100%)	0 (0%)	<0.001
	≥92.5	21 (45.65%)	25 (54.35%)	

## Data Availability

The original contributions presented in this study are included in the article/Appendix A. Further inquiries can be directed to the corresponding author.

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
