# Peer review of "Nomogram-Based Prediction of Survival in Stage IV Nasopharyngeal Carcinoma: A Retrospective Single-Center Study"

_diagnostics, 2025, doi:10.3390/diagnostics15111309_

Round 1

Reviewer 1 Report

Comments and Suggestions for Authors

This study presents a clinically meaningful attempt to develop a prognostic model for advanced nasopharyngeal carcinoma (NPC) based on pre- and post-treatment BMI and SIRI. Focusing specifically on NPC, which is relatively prevalent in the authors’ region, is reasonable and makes the findings particularly relevant in endemic areas. The retrospective single-institution design was likely possible due to the high case volume in such regions, but it also inevitably limits the generalizability of the findings. Nonetheless, the study provides valuable insights for NPC management. I encourage the authors to address the following points to further strengthen the manuscript prior to publication.

Major Comments

1, Clarification Needed Regarding Reverse Causality and Tumor Status

The manuscript treats low posttreatment BMI and substantial BMI reduction as independent prognostic factors under the assumption that they are causally linked to poor outcomes. While the Results section provides recurrence and persistence data (e.g., disease persistence in 45.9% of patients), the possibility that BMI decline may be a result of residual or progressive disease (i.e., cancer-related cachexia) is not directly discussed. This raises the issue of reverse causality, which should be acknowledged more explicitly in the Discussion.

If possible, subgroup analyses comparing BMI change among patients with and without recurrence/persistence would help clarify whether BMI loss is simply a marker of disease activity. Alternatively, a discussion acknowledging this limitation and its potential impact on the interpretation of BMI as a prognostic factor would strengthen the manuscript.

2, SIRI Cutoff and Disease Burden

Similar to Comment 1, the interpretation of pretreatment SIRI ≥125 as an independent poor prognostic marker should be discussed in relation to disease burden. For instance, bulky tumors that are not cured by CRT may inherently present with elevated SIRI levels. Moreover, systemic inflammation may be confounded by complications such as infections (e.g., pneumonia or sinusitis), particularly in the post-treatment setting. Lastly, the cutoff value derived from a single-institution cohort may not be generalizable, and this limitation should be more clearly stated.

3, Intervention Potential of the Predictive Parameters

If the authors aim to emphasize the clinical utility of BMI and SIRI, they should clarify whether these are viewed as modifiable risk factors. For example, is nutritional support expected to improve survival, even in patients unlikely to achieve CR? Is BMI simply a surrogate for overall resilience to therapy? The current discussion (e.g., lines 250–257, 266) alludes to their accessibility but does not clarify their potential as targets for clinical intervention. Explicit discussion is necessary to clarify the clinical implications of identifying these factors.

4, Lack of Discussion on Discrepancies with Prior Studies

Previous studies have emphasized pretreatment BMI as a stronger prognostic indicator than posttreatment BMI. This discrepancy is neither acknowledged nor discussed. Differences in timing of BMI measurement, degree of nutritional support, or institutional practices may explain this divergence. The authors should critically discuss why posttreatment BMI was more predictive in their study and whether this is unique to NPC or their cohort.

5, Overstatement of the Nomogram’s Clinical Utility

While the nomogram was reasonably constructed, its predictive strength may be overstated. The internal c-index was only 0.69, and no external validation was performed. Statements such as “strong ability to differentiate…” (e.g., L.236) should be tempered to reflect the modest discriminatory performance and lack of external validation. This is particularly important since the nomogram is presented as a potential clinical tool.

6, Overgeneralization of the Significance of ΔBMI

The variable “change in BMI” was significantly associated only with DFS, not OS or DSS. However, in several places (e.g., L.249–250), it is grouped with posttreatment BMI and SIRI as if all three were similarly associated with overall survival. This is misleading. The authors should clearly state that ΔBMI was only predictive of DFS and avoid conflating it with survival in general.

Author Response

To The Reviewer 1:

Thank you very much for taking the time to review this manuscript (diagnostics-3640065). Please find the detailed responses below and the corresponding revisions/corrections highlighted/in track changes in the re-submitted files

Comment 1: Clarification Needed Regarding Reverse Causality and Tumor Status

The manuscript treats low posttreatment BMI and substantial BMI reduction as independent prognostic factors under the assumption that they are causally linked to poor outcomes. While the Results section provides recurrence and persistence data (e.g., disease persistence in 45.9% of patients), the possibility that BMI decline may be a result of residual or progressive disease (i.e., cancer-related cachexia) is not directly discussed. This raises the issue of reverse causality, which should be acknowledged more explicitly in the Discussion.

If possible, subgroup analyses comparing BMI change among patients with and without recurrence/persistence would help clarify whether BMI loss is simply a marker of disease activity. Alternatively, a discussion acknowledging this limitation and its potential impact on the interpretation of BMI as a prognostic factor would strengthen the manuscript.

Response 1:

Thank you for the suggestion. We conducted subgroup analyses comparing both the change in BMI and posttreatment BMI among patients with and without recurrence or persistence. Patients with recurrence or persistence had a higher proportion of individuals with a BMI change of < –1.93 kg/m² and a posttreatment BMI of < 21.6 kg/m² compared to those without recurrence or persistence (Supplementary Table S2). We acknowledge that reverse causality may be a contributing factor in these findings, and we have revised the manuscript to more clearly discuss this as a potential limitation.

Revised text:

  • Subgroup analyses were conducted to compare changes in BMI, posttreatment BMI, and pretreatment SIRI between patients with and without recurrence or persistent disease. Patients experiencing recurrence or persistence had a higher proportion with a change in BMI of < -1.93 kg/m², posttreatment BMI < 21.6 kg/m², and pretreatment SIRI ≥ 125, compared to those without (Supplementary Table S2). These findings suggest that patients with residual or recurrent disease may experience more severe cancer-related cachexia or heightened systemic inflammation. However, the potential for reverse causality in this context should be considered. (page 11, line 342-349)

Comment 2: SIRI Cutoff and Disease Burden

Similar to Comment 1, the interpretation of pretreatment SIRI ≥125 as an independent poor prognostic marker should be discussed in relation to disease burden. For instance, bulky tumors that are not cured by CRT may inherently present with elevated SIRI levels. Moreover, systemic inflammation may be confounded by complications such as infections (e.g., pneumonia or sinusitis), particularly in the post-treatment setting. Lastly, the cutoff value derived from a single-institution cohort may not be generalizable, and this limitation should be more clearly stated.

Response 2:

Thank you for the comment. We also performed subgroup analyses of posttreatment SIRI among patients with and without recurrence or persistence. Patients with recurrence or persistence had a higher proportion of individuals with a pretreatment SIRI ≥125 compared to those without recurrence or persistence (Supplementary table S2). We have revised the manuscript to highlight the potential for reverse causality related to pretreatment SIRI. Additionally, we have addressed the confounding factors associated with SIRI and the limitations regarding the applicability of our cutoff values.

Revised text:

  • Subgroup analyses were conducted to compare changes in BMI, posttreatment BMI, and pretreatment SIRI between patients with and without recurrence or persistent disease. Patients experiencing recurrence or persistence had a higher proportion with a change in BMI of < -1.93 kg/m², posttreatment BMI < 21.6 kg/m², and pretreatment SIRI ≥ 125, compared to those without (Supplementary Table S2). These findings suggest that patients with residual or recurrent disease may experience more severe cancer-related cachexia or heightened systemic inflammation. However, the potential for reverse causality in this context should be considered. (page 11, line 342-349)
  • The cutoff values derived from this single-institution cohort may have limited generalizability to other populations. (page 11, line 381-382)
  • Fifth, systemic inflammation markers such as posttreatment SII and SIRI may be influenced by confounding factors, including infections. (page 11, line 383-384)

Comment 3: Intervention Potential of the Predictive Parameters

If the authors aim to emphasize the clinical utility of BMI and SIRI, they should clarify whether these are viewed as modifiable risk factors. For example, is nutritional support expected to improve survival, even in patients unlikely to achieve CR? Is BMI simply a surrogate for overall resilience to therapy? The current discussion (e.g., lines 250–257, 266) alludes to their accessibility but does not clarify their potential as targets for clinical intervention. Explicit discussion is necessary to clarify the clinical implications of identifying these factors.

Response 3:

Thank you for the comment. The ultimate goal of our investigation was to develop a nomogram for survival prediction. BMI and SIRI were included as indicators of patients’ nutritional and inflammatory status, respectively.

Our study identified posttreatment BMI as a predictive factor. Nutritional status is complex and may vary with age and the timing of measurement. Previous studies have shown that weight loss tends to peak during the middle to late phases of CCRT, underscoring the importance of active surveillance and timely intervention. However, cancer cachexia is a multifactorial syndrome resulting from metabolic, immunological, and endocrinological dysfunctions. Nutritional supplementation alone may not be sufficient to reverse cachexia. Further research is needed to determine whether improving nutritional status can lead to increased BMI and ultimately better survival outcomes.

SIRI serves as a composite marker of chronic low-grade inflammation, incorporating monocyte, neutrophil, and lymphocyte counts. These immune cells play complex and sometimes opposing roles in tumor progression and immune response. Whether SIRI is modifiable remains uncertain. Thus, the optimal strategies to influence it in a clinically beneficial way remain unclear.

We have revised the discussion to suggest that BMI may be modifiable risk factors, while emphasizing that further investigation is needed to establish whether interventions targeting nutritional and inflammatory status can improve survival outcomes.

Revised text:

  • The observed decline in BMI and elevated SIRI may reflect treatment-induced cachexia, poor recovery, and overall cancer burden. Nutritional status is a multifactorial variable influenced by age and the timing of measurement [42]. Previous studies have shown that weight loss typically peaks during the middle to late phases of CCRT [41], underscoring the importance of active surveillance and timely intervention. However, cancer cachexia is a complex syndrome resulting from metabolic, immunological, and endocrine dysregulation, and nutritional supplementation alone is often insufficient to reverse it [43]. Therefore, the notion that improving nutritional status—and consequently increasing posttreatment BMI—can directly enhance cancer survival may be overly simplistic. (page 10, line 325-334)
  • SIRI, which incorporates monocyte, neutrophil, and lymphocyte counts, serves as an integrated marker of chronic low-grade inflammation. These immune cells play complex and interrelated roles in both pro- and antitumor immunity. Whether SIRI is modifiable remains uncertain. In this study, changes in BMI, posttreatment BMI, and pretreatment SIRI appear to reflect a patient’s overall resilience to therapy. Further research is needed to clarify whether modifying nutritional and inflammatory statuses can translate into improved survival outcomes. (page 11, line 335-341)

Comment 4: Lack of Discussion on Discrepancies with Prior Studies

Previous studies have emphasized pretreatment BMI as a stronger prognostic indicator than posttreatment BMI. This discrepancy is neither acknowledged nor discussed. Differences in timing of BMI measurement, degree of nutritional support, or institutional practices may explain this divergence. The authors should critically discuss why posttreatment BMI was more predictive in their study and whether this is unique to NPC or their cohort.

Response 4:

We agree that variations in cancer type, timing of BMI measurement, and nutritional support practices may contribute to differences in study findings. Nonetheless, previous studies have demonstrated that nutritional status at diagnosis is a significant predictor of prognosis in patients with head and neck cancer. In our study, posttreatment BMI emerged as a unique prognostic factor in patients with locally advanced NPC. This may be attributed to the fact that maximal weight loss typically occurs during the mid to late stages of CCRT, with the greatest cumulative nutritional deficit observed in the posttreatment period. As such, posttreatment BMI may more accurately reflect the patient’s overall nutritional depletion and its impact on survival in this specific population. We have revised the discussion to incorporate this explanation.

Revised text:

  • We found that posttreatment BMI is associated with survival in patients with stage IV NPC, in contrast to the commonly reported association of pretreatment BMI with survival in other HNCs [15, 41]. Patients with HNC often experience swallowing difficulties due to the nature of the disease's location. Obstruction of the enteral route by advanced oral, oropharyngeal, or hypopharyngeal cancers lead to malnutrition prior to treatment. In contrast, the most significant weight loss typically occurs during the middle to late stages of CCRT in NPC, coinciding with an increased risk of malnutrition in the later phases of treatment. This suggests that the greatest cumulative nutritional loss occurs after treatment completion, likely due to treatment-related toxicities. However, previous studies have not explored the relationship between posttreatment nutritional status and survival in locally advanced NPC. Our findings build upon earlier research, demonstrating that posttreatment BMI is a distinct and meaningful prognostic factor in this context. Nevertheless, caution is warranted, as BMI may be influenced by cancer type, the timing of its assessment, and variations in nutritional support practices. (page 10, line 311-324)

Comment 5: Overstatement of the Nomogram’s Clinical Utility

While the nomogram was reasonably constructed, its predictive strength may be overstated. The internal c-index was only 0.69, and no external validation was performed. Statements such as “strong ability to differentiate…” (e.g., L.236) should be tempered to reflect the modest discriminatory performance and lack of external validation. This is particularly important since the nomogram is presented as a potential clinical tool.

Response 5:

Thank you for your suggestion. We agree that the term “strong” may have been overstated. We have revised the manuscript to adopt a more measured tone.

Revised text:

  • The nomogram serves as a tool to differentiate between patients with disease persistence or recurrence and those who remained disease-free (Table 4). (page 8, line 246-247)

Comment 6: Overgeneralization of the Significance of ΔBMI

The variable “change in BMI” was significantly associated only with DFS, not OS or DSS. However, in several places (e.g., L.249–250), it is grouped with posttreatment BMI and SIRI as if all three were similarly associated with overall survival. This is misleading. The authors should clearly state that ΔBMI was only predictive of DFS and avoid conflating it with survival in general.

Response 6:

Thank you for your thoughtful reminder. We have revised both the abstract and the manuscript to clarify the individual associations of posttreatment BMI, change in BMI, and pretreatment SIRI with OS, DSS, and DFS.

Revised abstract:

  • Conclusions: Associations with survival were observed between posttreatment BMI and OS, DSS, and DFS; pretreatment SIRI and DSS/DFS; and changes in BMI and DFS among patients with stage IV NPC. The developed nomogram aids in survival prediction. (page 1, line 38-41)

Revised text:

  • In this study, we identified significant associations between survival outcomes and several factors: posttreatment BMI was linked to OS, DSS, and DFS; pretreatment SIRI was associated with DSS and DFS; and changes in BMI were related to DFS. These factors serve as independent prognostic parameters of 5-year survival in patients with stage IV NPC. (page 9, line 258-261)
  • Associations with survival were observed between posttreatment BMI and OS, DSS, and DFS; pretreatment SIRI and DSS/DFS; and changes in BMI and DFS among patients with stage IV NPC. (page 12, 398-400)

Reviewer 2 Report

Comments and Suggestions for Authors

This manuscript presents a retrospective cohort study evaluating prognostic factors in patients with stage IVA nasopharyngeal carcinoma (NPC), with a particular focus on the clinical significance of posttreatment BMI, change in BMI, and pretreatment systemic inflammatory response index (SIRI). The authors also develop a novel nomogram incorporating these markers to predict disease-specific survival (DSS) outcomes.

A few minor revisions are necessary to improve clarity.

-The title could be more specific. Consider revising to: “Nomogram-Based Prediction of Survival in Stage IV Nasopharyngeal Carcinoma: A Retrospective Single-Center Study

-In the Introduction, consider adding linking sentences to connect sections (e.g., from epidemiological data to etiological factors, and from treatment standards to unmet prognostic needs).

-Briefly state why SIRI and BMI were prioritized over other indices; whether due to prior studies, biological relevance, or preliminary findings.

-The terms "head and neck cancers (HNCs)" and "NPC" are used somewhat interchangeably in the latter part of the introduction. Ensure consistent focus on NPC unless referencing broader HNC data with clear justification.

-The inclusion criteria are clear, but the exclusion criteria are not explicitly listed, which limits reproducibility. Include clear exclusion criteria (e.g., incomplete records, loss to follow-up, etc.) to improve methodological transparency.

-Please clarify whether staging was retrospectively updated for earlier cases.

-Please include additional variables in Table 1, such as smoking status, EBV status, performance status, and comorbidity index if available.

-The discussion claims novelty in posttreatment BMI being more predictive than pretreatment BMI, but does not explore why this difference may exist. Provide potential biological or treatment-related reasons for this divergence (e.g., treatment-induced cachexia, poor recovery capacity). Discuss whether BMI was adjusted for muscle mass or sarcopenia.

-Integrate a more cohesive explanation of how systemic inflammation (via neutrophils, monocytes) undermines immune surveillance in NPC specifically, referencing EBV-related immunosuppression where relevant.

-Discuss more analytically how the tool could actually change patient management e.g., early nutritional intervention, immunotherapy consideration, trial eligibility.

-Discuss whether gender was tested as a covariate in the model, and how its imbalance might skew generalizability or biomarker performance.

-Please, soften the conclusion to highlight the potential clinical relevance of the findings, while emphasizing the need for external validation and confirmation through prospective studies.

Comments on the Quality of English Language

editing for grammar and syntax

Author Response

To The Reviewer 2:

Thank you very much for taking the time to review this manuscript (diagnostics-3640065). Please find the detailed responses below and the corresponding revisions/corrections highlighted/in track changes in the re-submitted files.

Comment 1:

The title could be more specific. Consider revising to: “Nomogram-Based Prediction of Survival in Stage IV Nasopharyngeal Carcinoma: A Retrospective Single-Center Study”

Response 1:

Thank you for the suggestion. We have revised the title according to your suggestion.

Revised title:

Nomogram-Based Prediction of Survival in Stage IV Nasopharyngeal Carcinoma: A Retrospective Single-Center Study

Comment 2:

In the Introduction, consider adding linking sentences to connect sections (e.g., from epidemiological data to etiological factors, and from treatment standards to unmet prognostic needs).

Response 2:

Thank you for the comment. We have revised the introduction and added transitional sentences to improve the flow and better connect the sections.

Revised text:

  • Given the biological characteristics of NPC, including its high sensitivity to irradiation, effective disease management depends heavily on accurate staging and appropriate treatment planning. (page 2, line 67-69)
  • This is particularly relevant in advanced NPC, where therapeutic outcomes are closely tied to the patient’s response to chemoradiation. (page 2, line 82-84)

Comment 3:

Briefly state why SIRI and BMI were prioritized over other indices; whether due to prior studies, biological relevance, or preliminary findings.

Response 3:

SIRI and BMI were prioritized based on preliminary findings and their biological relevance to treatment resilience. Variables found to be statistically significant in univariate analyses were subsequently included in the multivariate models, which further identified SIRI and BMI as independent predictors. Both markers serve as surrogates for a patient’s overall resilience to therapy. We have revised the discussion to clarify these points.

Revised text:

  • In this study, changes in BMI, posttreatment BMI, and pretreatment SIRI appear to reflect a patient’s overall resilience to therapy. (page 11, line 338-339)

Comment 4:

The terms "head and neck cancers (HNCs)" and "NPC" are used somewhat interchangeably in the latter part of the introduction. Ensure consistent focus on NPC unless referencing broader HNC data with clear justification.

Response 4:

Thank you for the reminder. The term “HNC” used in the latter part of introduction was intended to mention HNCs and not NPCs. HNC appeared twice in the introduction referencing broader HNC data.

Comment 5:

The inclusion criteria are clear, but the exclusion criteria are not explicitly listed, which limits reproducibility. Include clear exclusion criteria (e.g., incomplete records, loss to follow-up, etc.) to improve methodological transparency.

Response 5:

Thank you for the comment. We have included the exclusion criteria in the “Patients and data acquisition” section.

Revised text:

  • The patient exclusion criteria were as follows: (1) evidence of a distal metastasis at diagnosis; (2) treated with modalities other than CCRT with/without IC; (3) incomplete pre-treatment and posttreatment medical records; (4) patients lost to follow-up. (page 3, line 133-136)

Comment 6:

Please clarify whether staging was retrospectively updated for earlier cases.

Response 6:

Earlier cases were retrospectively re-staged according to the 8th edition of the AJCC staging system. We have revised the description to improve clarity.

Revised text:

  • Radiologists reviewed the computed tomography scan or magnetic resonance imaging results, and the patients were retrospectively re-staged according to the 8th edition of the AJCC staging manual [10]. (page 3, line 119-121)

Comment 7:

Please include additional variables in Table 1, such as smoking status, EBV status, performance status, and comorbidity index if available.

Response 7:

Thank you for the comment. We have added the smoking status, performance status, and comorbidity index of our patients to Table 1. We apologize for the incomplete EBV status data, as EBV-DNA testing was not routinely implemented at our hospital until 2016.

Revised table 1:

Table 1. Characteristics and clinicopathological details of the recruited patients

N (%) or Mean±SD or median (range,IQR)

Age (years)

56.7±11 (24-88)

Gender

Male

55 (90.16%)

Female

6 (9.84%)

Clinical T Classification

T1/T2/T3/T4

12(19.67%)/7(18.03%)/12(19.67%)/30(49.18%)

Clinical N Classification

N0/N1/N2/N3

5(8.20%)/12(19.67%)/9(14.75%)/20(32.79%)

Body Height (cm)

166.7±7.4 (149-189)

Pretreatment BW (kg)

69.6±16.5 (43-146)

Posttreatment BW (kg)

62.2±11.9 (41-107)

Pretreatment BMI (kg/m2)

24.9±4.5 (16.38-40.87)

Posttreatment BMI (kg/m2)

62.2±11.9 (41-107)

Induction chemotherapy

With/Without

37 (60.66%)/24 (39.34%)

Death within 5 years

28 (45.90%)

Disease Persistence

28 (45.90%)

Recurrence

Local/Regional/Distant

12 (19.67%)/9 (14.75%)/24 (39.34%)

Follow-up duration (days)

1716±1096 (169-5281)

Smoking status

Yes/No

34 (55.74%)/27 (44.26%)

ECOG performance status

0/1/2/4

30 (49.18%)/29 (47.54%)/1 (1.64%)/1 (1.64%)

Hypertension

Yes/No

5 (8.20%)/56 (91.80%)

Diabetes

Yes/No

2 (3.28%)/59 (96.72%)

Hepatitis B

Yes/No

3 (4.92%)/58 (95.08%)

Coronary artery disease

Yes/No

2 (3.28%)/59 (96.72%)

BW: body weight; BMI: body mass index; ECOG: Eastern Cooperative Oncology Group

Comment 8:

The discussion claims novelty in posttreatment BMI being more predictive than pretreatment BMI, but does not explore why this difference may exist. Provide potential biological or treatment-related reasons for this divergence (e.g., treatment-induced cachexia, poor recovery capacity). Discuss whether BMI was adjusted for muscle mass or sarcopenia.

Response 8:

Thank you for the comment. Previous studies have highlighted pretreatment BMI as a stronger prognostic indicator compared to posttreatment BMI. In contrast, posttreatment BMI is a unique prognostic factor observed in our cohort. Differences in cancer type, timing of BMI measurement, and nutritional support practices may contribute to this discrepancy. Maximal weight loss typically occurs during the middle to late stages of CCRT in patients with locally advanced NPC. The greatest cumulative nutritional deficit is observed after treatment completion. As such, posttreatment BMI is particularly relevant to survival outcomes in this patient group. We have revised the discussion to incorporate this explanation.

Revised text:

  • We found that posttreatment BMI is associated with survival in patients with stage IV NPC, in contrast to the commonly reported association of pretreatment BMI with outcomes in other HNCs [15, 41]. Patients with HNC often experience swallowing difficulties due to the nature of the disease's location. Obstruction of the enteral route by advanced oral, oropharyngeal, or hypopharyngeal cancers lead to malnutrition prior to treatment. In contrast, the most significant weight loss typically occurs during the middle to late stages of CCRT in NPC, coinciding with an increased risk of malnutrition in the later phases of treatment. This suggests that the greatest cumulative nutritional loss occurs after treatment completion, likely due to treatment-related toxicities. However, previous studies have not explored the relationship between posttreatment nutritional status and survival in locally advanced NPC. Our findings build upon earlier research, demonstrating that posttreatment BMI is a distinct and meaningful prognostic factor in this context. Nevertheless, caution is warranted, as BMI may be influenced by cancer type, the timing of its assessment, and variations in nutritional support practices. (page 10, line 311-324)

We acknowledge that BMI is a crude surrogate of nutritional status, and that other indicators, such as sarcopenia or muscle mass, may provide a more accurate reflection of a patient's nutritional state. We have included this as a limitation in the manuscript.

Revised text:

  • Additionally, BMI is a crude indicator of nutritional status; incorporating measures such as muscle mass or assessing for sarcopenia would provide a more accurate representation of a patient’s nutritional condition. (page 11-12, line 384-387)

Comment 9:

Integrate a more cohesive explanation of how systemic inflammation (via neutrophils, monocytes) undermines immune surveillance in NPC specifically, referencing EBV-related immunosuppression where relevant.

Response 9:

The precise mechanism underlying the prognostic role of SIRI in NPC and its relationship with EBV remains unclear. However, the prognostic significance of SIRI can be partially explained by the functions of its constituent cells. Neutrophils contribute to immunosuppression by releasing cytokines such as interleukin-10 (IL-10) and transforming growth factor-β (TGF-β), which inhibit T cell proliferation and activation. In contrast, lymphocytes exert antitumor effects by suppressing tumor growth and metastasis through the secretion of interferon-γ (IFN-γ) and tumor necrosis factor-α (TNF-α). Monocytes and monocyte-derived M2 macrophages facilitate tumor progression by promoting growth, invasion, immune evasion, and metastasis. We have added a paragraph in the discussion to elaborate on this explanation.

Revised text:

  • The underlying mechanism linking SIRI to prognosis in NPC, as well as its relationship with EBV, remains unclear. However, the prognostic value of SIRI may be attributed to its individual components. Neutrophils can promote immunosuppression by releasing cytokines such as interleukin-10 (IL-10) and transforming growth factor-β (TGF-β), which inhibit T cell proliferation and activation. In contrast, lymphocytes have antitumor activity, suppressing tumor growth and metastasis through the secretion of interferon-γ (IFN-γ) and tumor necrosis factor-α (TNF-α). Monocytes and monocyte-derived M2 macrophages contribute to tumor progression by facilitating growth, invasion, immune evasion, and metastasis [40]. (page 10, line 302-310)

Comment 10:

Discuss more analytically how the tool could actually change patient management e.g., early nutritional intervention, immunotherapy consideration, trial eligibility.

Response 10:

The nomogram serves as an analytical tool for patient stratification. Patients with a nomogram score of < 92.5 had a 2.2-fold increased risk of disease recurrence compared to those with a score of ≥ 92.5 (100% versus 45.65%, p = 0.001). Patients identified as having a poor prognosis may benefit from more intensive nutritional support. Physicians should consider early enrollment of these patients in clinical trials rather than relying solely on conventional therapies. We have revised the discussion to provide a more detailed explanation of the nomogram's functionality.

Revised text:

  • Patients with poor prognosis may be appropriate candidates for intensive nutritional support. Physicians should consider early enrollment of these individuals in clinical trials as an alternative to standard therapies. (page 11, line 367-369)

Comment 11:

Discuss whether gender was tested as a covariate in the model, and how its imbalance might skew generalizability or biomarker performance.

Response 11:

The multivariate analyses were conducted using a forward stepwise Cox regression model adjusted for age and gender. Gender was included as a covariate in the multivariate analyses and was not an independent risk factor for OS (HR 1.710, 95% CI 0.400–7.560, p=0.460), DSS (HR 1.527, 95% CI 0.349–6.674, p=0.574), or DFS (HR 3.345, 95% CI 0.934–11.982, p=0.064). The high male-to-female ratio in our cohort is acknowledged as a limitation. This demographic difference may result in a cohort that is not fully representative of other populations, thus limiting the generalizability of our findings and the performance of the biomarkers. We have revised the “Statistical Analysis” section for clarification.

Reference to text:

  • Finally, the high male-to-female ratio in our cohort may differ significantly from other populations. Based on data from the Taiwan Cancer Registry Database, the nationwide male-to-female ratio for stage IV NPC without metastasis ranged from 2.67 to 3.94 between 2017 and 2021. The high male-to-female ratio reflects the demographics of local stage IV NPC patients. (page 12, line 387-391)

Revised text:

  • Variables that were statistically significant (p < 0.05) in univariate analyses were included in multivariate Cox regression analyses, adjusted for age and gender, using a forward stepwise method to address multicollinearity among parameters. (page 4, line 164-166)

Comment 12:

Please, soften the conclusion to highlight the potential clinical relevance of the findings, while emphasizing the need for external validation and confirmation through prospective studies.

Response 12:

Thank you for the suggestion. We have removed the sentence, “These parameters should be routinely assessed in advanced NPC patients,” to present a more measured conclusion. The need for external validation and further confirmation has been reiterated in the conclusion.

Revised text:

  • Our developed nomogram may aid in the stratification of patients and individualized treatment strategy selection. Additional external validation and confirmation through large cohort prospective studies is necessary. (page 12, line 400-403)

Response to Comments on the Quality of English Language

Comment by reviewer: editing for grammar and syntax

Response:

Thank you for your suggestion. We have carefully re-read and revised the manuscript, correcting several grammatical and syntactical errors.

Round 2

Reviewer 1 Report

Comments and Suggestions for Authors

Dear Authors,

Thank you very much for your efforts in revising the manuscript. I appreciate your responsiveness to the previous comments, and I acknowledge the improvements made. However, several important issues remain insufficiently addressed and require further clarification or revision. Please find below my specific observations and suggestions for improvement.

  1. Reverse Causality and Tumor Status (Comment 1)

Your revision:
You added subgroup analyses and acknowledged the possibility of reverse causation between low BMI and disease recurrence or persistence.

Remaining concern:
The additional data are welcome; however, the current discussion still suggests that low posttreatment BMI and BMI reduction are independent prognostic factors, without adequately emphasizing that these markers may simply reflect ongoing disease activity.

For instance, the following statement:

“These findings suggest that patients with residual or recurrent disease may experience more severe cancer-related cachexia…”

implies a directional relationship (disease → BMI loss) while also treating BMI as a prognostic predictor, which creates a logical inconsistency. Furthermore, recurrence or persistence was not included as a covariate in the Cox regression model, raising concerns that BMI may be acting as a proxy for underlying tumor activity, rather than as an independent predictor.

Suggested revision:
Please revise the discussion to clarify that BMI decline may result from persistent or recurrent disease, and therefore may not serve as an independent predictor. For example:

“These findings may reflect that patients with residual or recurrent disease have lower BMI and greater weight loss as a result of cancer progression or cachexia, rather than these being independent predictors of prognosis. Therefore, the possibility of reverse causation should be carefully considered in interpreting these variables as prognostic factors.”

Alternatively:

“These findings suggest that low posttreatment BMI and greater BMI decline may reflect the presence of persistent or progressive disease, potentially as a consequence of cancer-related cachexia. As such, these factors should be interpreted with caution, considering the potential influence of reverse causation.”

Additionally, presenting Kaplan-Meier curves stratified by BMI levels within the non-recurrent group would help assess whether low BMI retains prognostic value independent of recurrence.

  1. Interpretation of SIRI and Potential Confounding (Comment 2)

Your revision:
You acknowledged potential reverse causality, infectious confounders, and the limited generalizability of the cutoff value.

Remaining concern:
Despite these additions, the discussion still treats SIRI as a meaningful pretreatment predictor without addressing the possibility that elevated SIRI may reflect tumor bulk, unresectability, or EBV-driven inflammation. Furthermore, the multivariable model did not include key clinical confounders such as EBV DNA levels, T/N staging, or tumor volume, which may lead to overestimation of SIRI’s independent predictive value.

Suggested revision:
Please clarify that SIRI may reflect tumor burden or systemic inflammation rather than function as a modifiable or truly independent factor. Consider the following wording:

“Although SIRI was associated with survival outcomes in our study, it may also reflect underlying tumor burden or EBV-related inflammation rather than acting as an independent predictor. Notably, key prognostic factors such as EBV DNA levels or detailed T classification were not included in the current multivariable model, which may confound the observed association. Therefore, interpretation of SIRI should be made with caution, and further studies are warranted to determine its independent prognostic value.”

Alternatively:

“Although pretreatment SIRI showed significant association with prognosis, its role as an independent predictor remains uncertain. Elevated SIRI may be closely linked to tumor burden, EBV activity, or the inflammatory response to bulky disease, and could be confounded by clinical factors such as T stage or EBV DNA levels. Including these established prognostic markers in multivariable analysis would help clarify whether SIRI offers additive predictive value. A sensitivity analysis incorporating such variables is warranted.”

  1. Variable Selection in the Nomogram (Related to Comment 5)

Your revision:
You appropriately softened the language regarding the nomogram’s predictive strength.

Remaining concern:
The rationale for selecting only two variables (posttreatment BMI and pretreatment SIRI) in the final model is not clearly explained. It is unclear whether all significant univariate predictors were considered, and whether issues such as multicollinearity or model parsimony influenced the decision.

Suggested revision:
Please provide a clear description of the variable selection process and explain the reasoning behind the inclusion of only these two variables. Indicating whether multicollinearity or model simplification was a factor would enhance transparency.

  1. Omission of Established Prognostic Factors

Missing content:
Key clinical variables such as EBV DNA levels, ECOG performance status, and detailed T/N staging were not included in the multivariable model or the nomogram.

Concern:
These are widely recognized as strong independent prognostic factors in nasopharyngeal carcinoma. Their exclusion may lead to model misspecification and overemphasis on BMI and SIRI.

Suggested revision:
Please explicitly explain the exclusion of these variables. If these data were unavailable or incomplete, this should be clearly stated. If available, consider including them in the model or conducting a sensitivity analysis.

  1. Method for Determining Cutoff Values

Your revision:
You indicated that ROC analysis was used to define cutoff values.

Remaining concern:
It remains unclear whether cutoff values were selected independently of the outcome variables and whether potential overfitting was addressed. The method used (e.g., Youden index, time-dependent ROC) should be specified.

Suggested revision:
Please clarify the method and timing of cutoff selection. For example:

“Cutoff values for BMI and SIRI were determined using the Youden index derived from ROC analysis for 5-year DSS. These thresholds were defined prior to multivariable analysis to avoid data leakage.”

Addressing these points would improve the transparency, clarity, and clinical relevance of the manuscript. I hope these suggestions will help further refine and strengthen your work.

Sincerely,

Author Response

To The Reviewer 1:

Thank you again for taking the time to review this manuscript (diagnostics-3640065). Please find the detailed responses below and the corresponding revisions/corrections highlighted/in track changes in the re-submitted files

Comment 1: Reverse Causality and Tumor Status (Comment 1)

Your revision:

You added subgroup analyses and acknowledged the possibility of reverse causation between low BMI and disease recurrence or persistence.

Remaining concern:

The additional data are welcome; however, the current discussion still suggests that low posttreatment BMI and BMI reduction are independent prognostic factors, without adequately emphasizing that these markers may simply reflect ongoing disease activity.

For instance, the following statement:

“These findings suggest that patients with residual or recurrent disease may experience more severe cancer-related cachexia…”

implies a directional relationship (disease → BMI loss) while also treating BMI as a prognostic predictor, which creates a logical inconsistency. Furthermore, recurrence or persistence was not included as a covariate in the Cox regression model, raising concerns that BMI may be acting as a proxy for underlying tumor activity, rather than as an independent predictor.

Suggested revision:

Please revise the discussion to clarify that BMI decline may result from persistent or recurrent disease, and therefore may not serve as an independent predictor. For example:

“These findings may reflect that patients with residual or recurrent disease have lower BMI and greater weight loss as a result of cancer progression or cachexia, rather than these being independent predictors of prognosis. Therefore, the possibility of reverse causation should be carefully considered in interpreting these variables as prognostic factors.”

Alternatively:

“These findings suggest that low posttreatment BMI and greater BMI decline may reflect the presence of persistent or progressive disease, potentially as a consequence of cancer-related cachexia. As such, these factors should be interpreted with caution, considering the potential influence of reverse causation.”

Additionally, presenting Kaplan-Meier curves stratified by BMI levels within the non-recurrent group would help assess whether low BMI retains prognostic value independent of recurrence.

Response 1:

Thank you for the suggestion. We understand and agree with your concern. We have revised the discussion according to your suggested revision.

Revised text:

  • These findings may reflect that patients with residual or recurrent disease have lower BMI and greater weight loss as a result of cancer progression or cachexia, rather than these being independent predictors of prognosis. Therefore, the possibility of reverse causation should be carefully considered in interpreting these variables as prognostic factors. (page 11, line 346-350)

In addition, we have included the Kaplan-Meier curves as supplementary “Figure S1” and “Figure S2” for your review. Posttreatment BMI retains prognostic value for DSS within the non-recurrent group, whereas change in BMI did not show prognostic significance in this subgroup. This supports our finding that posttreatment BMI is a prognostic indicator while change in BMI is not a prognostic indicator for DSS.

Figure S1:

Figure S2:

Revised supplementary materials:

Figure S1: The 1-year, 3-year, and 5-year DSS for stage IV NPC patients with posttreatment BMI ≥ 21.6 versus posttreatment BMI < 21.6 within the non-recurrent group were 100%, 100%, and 100% versus 78%, 78%, and 78%, respectively; Figure S2: The 1-year, 3-year, and 5-year DSS for stage IV NPC patients with Δ BMI ≥ -1.93 versus Δ BMI < -1.93 within the non-recurrent group were 94%, 94%, and 94% versus 92%, 92%, and 92%, respectively. (page 12, line 416-421)

Comment 2: Interpretation of SIRI and Potential Confounding (Comment 2)

Your revision:

You acknowledged potential reverse causality, infectious confounders, and the limited generalizability of the cutoff value.

Remaining concern:

Despite these additions, the discussion still treats SIRI as a meaningful pretreatment predictor without addressing the possibility that elevated SIRI may reflect tumor bulk, unresectability, or EBV-driven inflammation. Furthermore, the multivariable model did not include key clinical confounders such as EBV DNA levels, T/N staging, or tumor volume, which may lead to overestimation of SIRI’s independent predictive value.

Suggested revision:

Please clarify that SIRI may reflect tumor burden or systemic inflammation rather than function as a modifiable or truly independent factor. Consider the following wording:

“Although SIRI was associated with survival outcomes in our study, it may also reflect underlying tumor burden or EBV-related inflammation rather than acting as an independent predictor. Notably, key prognostic factors such as EBV DNA levels or detailed T classification were not included in the current multivariable model, which may confound the observed association. Therefore, interpretation of SIRI should be made with caution, and further studies are warranted to determine its independent prognostic value.”

Alternatively:

“Although pretreatment SIRI showed significant association with prognosis, its role as an independent predictor remains uncertain. Elevated SIRI may be closely linked to tumor burden, EBV activity, or the inflammatory response to bulky disease, and could be confounded by clinical factors such as T stage or EBV DNA levels. Including these established prognostic markers in multivariable analysis would help clarify whether SIRI offers additive predictive value. A sensitivity analysis incorporating such variables is warranted.”

Response 2:

Thank you for the suggestion. We agree with your concern and have revised the discussion according to your suggested revision.

Revised text:

  • Although SIRI was associated with survival outcomes in our study, it may also reflect underlying tumor burden or EBV-related inflammation rather than acting as an independent predictor. Notably, key prognostic factors such as EBV DNA levels or total tumor volume were not included in the current Cox regression analyses, which may confound the observed association. Therefore, interpretation of SIRI should be made with caution, and further studies are warranted to determine its independent prognostic value. (page 11, line 350-356)

Comment 3: Variable Selection in the Nomogram (Related to Comment 5)

Your revision:

You appropriately softened the language regarding the nomogram’s predictive strength.

Remaining concern:

The rationale for selecting only two variables (posttreatment BMI and pretreatment SIRI) in the final model is not clearly explained. It is unclear whether all significant univariate predictors were considered, and whether issues such as multicollinearity or model parsimony influenced the decision.

Suggested revision:

Please provide a clear description of the variable selection process and explain the reasoning behind the inclusion of only these two variables. Indicating whether multicollinearity or model simplification was a factor would enhance transparency.

Response 3:

Thank you for the comment. Variables that were statistically significant (p < 0.05) in univariate analyses were included in multivariate Cox regression analyses, adjusted for age and gender, using a forward stepwise method to address multicollinearity among parameters. Only variables that remained significant in the multivariate analysis are presented in Table 3. The variable selection process was described in “Statistical Analysis” section.

Reference to text:

  • Variables that were statistically significant (p < 0.05) in univariate analyses were included in multivariate Cox regression analyses, adjusted for age and gender, using a forward stepwise method to address multicollinearity among parameters.

Our nomogram was created using variables that were statistically significant (p < 0.05) in multivariate Cox regression analyses. We have revised the “Statistical Analysis” section to explain the reasoning behind the inclusion of only posttreatment BMI and pretreatment SIRI in our nomogram.

Revised text:

  • A nomogram was developed using variables that were statistically significant (p < 0.05) in multivariate Cox regression analyses to predict DSS following treatment completion in this patient cohort. (page 4, line 169-171)

Comment 4: Omission of Established Prognostic Factors

Missing content:

Key clinical variables such as EBV DNA levels, ECOG performance status, and detailed T/N staging were not included in the multivariable model or the nomogram.

Concern:

These are widely recognized as strong independent prognostic factors in nasopharyngeal carcinoma. Their exclusion may lead to model misspecification and overemphasis on BMI and SIRI.

Suggested revision:

Please explicitly explain the exclusion of these variables. If these data were unavailable or incomplete, this should be clearly stated. If available, consider including them in the model or conducting a sensitivity analysis.

Response 4:

We have included ECOG performance status in the univariate analysis and included the results in table 2.

Revised table 2:

Table 2. Univariate analyses of clinicopathological factors that related to overall survival (OS), disease-specific survival (DSS), and disease-free survival (DFS) in stage IV nasopharyngeal carcinoma patients

Variables

Dichotomized Units

OS

DSS

DFS

HR

95%CI

p-value

HR

95%CI

p-value

HR

95%CI

p-value

Gender

Female vs male

1.611

0.382

6.800

0.516

1.548

0.366

6.547

0.552

3.083

1.346

0.413

4.389

Age

<53 vs. ≥53

1.747

0.742

4.116

0.202

1.389

0.607

3.177

0.437

1.423

0.711

2.849

0.319

Pretreatment BMI (kg/m2)

<22.48 vs. ≥22.48

0.430

0.204

0.906

0.026

0.462

0.216

0.989

0.047

0.533

0.272

1.043

0.066

Posttreatment BMI (kg/m2)

<21.6 vs. ≥21.6

0.380

0.175

0.826

0.015

0.339

0.152

0.757

0.008

0.353

0.180

0.693

0.002

Δ BMI (kg/m2)

<-1.93 vs. ≥-1.93

0.634

0.292

1.378

0.250

0.570

0.255

1.272

0.170

0.443

0.213

0.921

0.029

T status

T1&T2 vs. T3&T4

2.880

0.996

8.328

0.051

0.275

0.948

7.980

0.063

1.657

0.776

3.536

0.192

N status

N0-1 vs. N2 vs. N3

0.681

0.445

1.019

0.062

0.653

0.433

0.985

0.042

0.723

0.502

1.040

0.080

Pretreatment Hb (gm/dl)

<16 vs. ≥16

0.513

0.194

1.353

0.117

0.638

0.220

1.851

0.408

0.800

0.311

2.060

0.644

Posttreatment Hb (gm/dl)

<11.2 vs. ≥11.2

0.582

0.276

1.225

0.154

0.724

0.339

1.544

0.403

0.948

0.488

1.842

0.875

Δ Hb (gm/dl)

<-2.9 vs. ≥-2.9

0.684

0.326

1.436

0.316

0.548

0.256

1.172

0.121

0.598

0.310

1.154

0.125

Pretreatment LMR

<3.6 vs. ≥3.6

0.456

0.210

0.990

0.047

0.410

0.184

0.914

0.029

0.553

0.286

1.070

0.078

Posttreatment LMR

<1.3 vs. ≥1.3

0.485

0.213

1.104

0.085

0.458

0.200

1.049

0.065

0.493

0.231

1.054

0.068

Δ LMR

<-2.98 vs. ≥-2.98

1.414

0.601

3.327

0.728

2.072

0.784

5.475

0.142

1.369

0.659

2.843

0.400

Pretreatment PLR

<1.3 vs. ≥1.3

1.748

0.741

4.210

0.202

1.661

0.701

3.938

0.249

1.362

0.680

2.728

0.383

Posttreatment PLR

<4 vs. ≥4

1.557

0.704

3.446

0.275

1.631

0.732

3.636

0.232

1.892

0.944

3.794

0.072

Δ PLR

<0.54 vs. ≥0.54

1.829

0.775

4.314

0.168

1.739

0.733

4.126

0.209

1.856

0.843

4.083

0.124

Pretreatment NLR

<2.8 vs. ≥2.8

2.575

1.134

5.851

0.024

2.953

1.248

6.988

0.014

2.725

1.355

5.481

0.005

Posttreatment NLR

<4.8 vs. ≥4.8

1.312

0.622

2.767

0.476

2.953

1.248

6.988

0.618

1.881

0.973

3.638

0.060

Δ NLR

<0.86 vs. ≥0.86

0.771

0.366

1.625

0.495

0.714

0.333

1.528

0.385

0.809

0.420

1.559

0.527

Pretreatment SII

<826 vs. ≥826

2.533

1.185

5.412

0.016

2.799

1.280

6.118

0.010

2.699

1.385

5.257

0.004

Posttreatment SII

<582 vs. ≥582

1.655

0.699

3.920

0.252

1.588

0.667

3.782

0.296

2.052

0.897

4.692

0.089

Δ SII

<372 vs. ≥372

0.340

0.103

1.127

0.078

0.352

0.106

1.170

0.089

0.636

0.277

1.457

0.284

Pretreatment SIRI

<125 vs. ≥125

2.260

1.056

4.838

0.036

2.935

1.315

6.552

0.009

2.558

1.297

5.044

0.007

Posttreatment SIRI

<124 vs. ≥124

0.804

0.382

1.691

0.565

0.751

0.352

1.600

0.458

0.980

0.501

1.916

0.953

Δ SIRI

<148 vs. ≥148

2.299

0.970

5.449

0.059

2.435

1.021

5.805

0.045

2.451

1.108

5.425

0.027

Induction chemotherapy

No vs. Yes

1.134

0.523

2.459

0.751

1.257

0.564

2.800

0.576

0.775

0.398

1.507

0.452

ECOG performance status

0-1 vs. 2-4

1.703

0.403

7.200

0.469

1.759

0.415

7.453

0.443

2.174

0.513

9.214

0.292

BH: body height; BW: body weight; BMI: body mass index; Hb: hemoglobin; Δ: change in the variable; LMR: lymphocyte/monocyte ratio; PLR: platelet/lymphocyte ratio; NLR: neutrophil/lymphocyte ratio; SII: systemic immune inflammation index; SIRI: systemic inflammation response index; ECOG: Eastern Cooperative Oncology Group

Variables found to be statistically significant in univariate analyses were subsequently included in the multivariate analyses using a forward stepwise Cox regression model adjusted for age and gender. Only variables with statistical significance (p<0.05) are shown in Table 3 for presentation purposes. T status did not show significance in univariate analysis; thus it was not included in the multivariate analysis. N status was statistically significant in DSS but did not remain significant in the multivariate analysis; therefore, it was excluded from Table 3. We have revised the legend of Table 3 for clarification.

Revised table 3:

Table 3. Multivariate logistic regression analyses of clinicopathological factors that related to overall survival (OS), disease-specific survival (DSS), and disease-free survival (DFS) in stage IV nasopharyngeal carcinoma patients (only presenting variables with statistical significance)

Variables

Dichotomized Units

OS

DSS

DFS

HR

95%CI

p-value

HR

95%CI

p-value

HR

95%CI

p-value

Posttreatment BMI (kg/m2)

<21.6 vs. ≥21.6

0.368

0.169

0.801

0.012

0.333

0.148

0.746

0.008

0.274

0.132

0.569

0.001

Δ BMI (kg/m2)

<-1.93 vs. ≥-1.93

0.268

0.118

0.609

0.002

Pretreatment SIRI

<125 vs. ≥125

2.841

1.256

6.429

0.012

3.541

1.717

7.304

0.001

BMI: body mass index; Δ: change in the variable; SIRI: systemic inflammation response index

We apologize for the incomplete EBV status data, as EBV-DNA testing was not routinely implemented at our hospital until 2016. We have included this as a limitation in the manuscript.

Revised text:

  • Moreover, the EBV status of our patients was incomplete, as EBV-DNA testing was not routinely implemented at our hospital until 2016. (page 11-12, line 385-386)

Comment 5: Method for Determining Cutoff Values

Your revision:

You indicated that ROC analysis was used to define cutoff values.

Remaining concern:

It remains unclear whether cutoff values were selected independently of the outcome variables and whether potential overfitting was addressed. The method used (e.g., Youden index, time-dependent ROC) should be specified.

Suggested revision:

Please clarify the method and timing of cutoff selection. For example:

“Cutoff values for BMI and SIRI were determined using the Youden index derived from ROC analysis for 5-year DSS. These thresholds were defined prior to multivariable analysis to avoid data leakage.”

Response 5:

Thank you for your suggestion. We have revised the “Statistical Analysis“ section to specify the method and timing of cutoff selection.

Revised text:

  • The optimal cutoff values for univariate analysis of continuous variables were determined using the Youden index derived from receiver operating characteristic (ROC) curve analysis, selecting the point with the highest accuracy for predicting recurrence or death. These thresholds were defined prior to multivariable analysis to avoid data leakage. (page 4, line 161-165)

Reviewer 2 Report

Comments and Suggestions for Authors

The authors covered the issues I mentioned.

Author Response

To The Reviewer 2:

Thank you so much for reviewing our manuscript (diagnostics-3640065).

Comment 1:

The authors covered the issues I mentioned.

Response 1:

Thank you for taking the time and effort to review our manuscript. We sincerely appreciate your insightful comments on our paper.

Round 3

Reviewer 1 Report

Comments and Suggestions for Authors

Thank you for your detailed and thoughtful responses to the previous review comments. Your revisions have significantly improved the clarity and rigor of the manuscript. In particular, the revised discussion around reverse causality, the addition of subgroup analyses (Figures S1 and S2), and the clarification of your statistical approach are highly appreciated.

I have no further major concerns. While a few minor issues—such as clarifying p-values for supplementary figures or explicitly noting reasons for the exclusion of certain variables—could enhance transparency, they do not materially impact the validity of your findings. I consider the manuscript suitable for publication in its current form or with only minor editorial adjustments.

Thank you again for your careful revisions.